



# Across-track Extension of Retrieved Cloud and Aerosol Properties for the EarthCARE Mission: The ACM-3D Product

Zhipeng Qu[1], Howard W. Barker[1], Jason N. S. Cole[1], Mark W. Shephard[1]

[1] Environment and Climate Change Canada, Toronto, ON, Canada

*Correspondence to*: Howard W. Barker (howard.barker@canada.ca)

**Abstract.** The narrow cross-section of cloud and aerosol properties retrieved by L2-algorithms that operate on data from EarthCARE's nadir-pointing sensors gets "broadened" across-track by an algorithm that is described and demonstrated here. This *Scene Construction Algorithm* (SCA) consists of four sub-algorithms. At its core is a radiance-matching procedure that works with measurements made by EarthCARE's Multi-Spectral Imager (MSI). In essence, an off-nadir pixel gets filled with retrieved profiles that are associated with a (nearby) nadir pixel whose MSI radiances best match those of the off-nadir pixel. The SCA constructs a 3D array of cloud and aerosol (and surface) properties for entire *frames* that measure ~6,000 km along-track by 150 km across-track (i.e., the MSI's full swath). Constructed domains out to ~15 km on both sides of nadir are used explicitly downstream as input for 3D radiative transfer models that predict top-of-atmosphere (TOA) broadband solar and thermal fluxes and radiances. These quantities are compared to commensurate measurements made by EarthCARE's BroadBand Radiometer (BBR), thus facilitating a continuous closure assessment of the retrievals. Three 6,000 km x 200 km frames of synthetic EarthCARE observations were used to demonstrate the SCA. The main conclusion is that errors in modelled TOA fluxes that stem from use of 3D domains produced by the SCA are expected to be less than $\pm 5$ W m$^{-2}$ and rarely larger than $\pm 10$ W m$^{-2}$. As such, the SCA, as purveyor of information needed to run 3D radiative transfer models, should help more than hinder the radiative closure assessment of EarthCARE's L2 retrievals.

## 1. Introduction

The objective of the EarthCARE satellite mission is to help improve numerical predictions of weather, air quality, and climatic change via application of synergistic L2-retrieval algorithms to observational data from its cloud-profiling radar (CPR), backscattering lidar (ATLID), and passive multi-spectral imager (MSI) (Illingworth et al. 2015). EarthCARE's overarching scientific goal (ESA 2001) is to retrieve cloud and aerosol properties with enough accuracy that when used to initialize atmospheric radiative transfer (RT) models, simulated top-of-atmosphere (TOA) broadband radiative fluxes, for domains covering ~100 km$^2$, agree with their observational-based counterparts to within $\pm 10$ W m$^{-2}$ *more often than not*. "Observed" TOA fluxes derive from radiances measured by EarthCARE's multi-angle broadband radiometer (BBR). As the latter are not used by retrieval algorithms, comparing them to modelled values, obtained



by RT models operating on retrieved quantities, affects a "moderately stringent" verification of the retrievals (Barker et al. 2023); "moderately stringent" because BBR radiances consist, in part, of photons that share the same gaseous pathlength and number of cloud-aerosol-surface scattering event distributions as those that constitute MSI radiances. These imperfections aside, this radiative closure verification is a well-defined and cost-effective final stage in Earth-

CARE's formal "production model" (Eisinger et al. 2023).

In light of EarthCARE's ambitious goal of limiting differences between measured and modelled TOA fluxes to $\pm 10$ W m$^{-2}$, the usefulness of its radiative closure programme depends much on reducing errors and uncertainties in: BBR measurements; variables needed by RT models that are not provided by EarthCARE observations; and RT models. Included in this are issues of observational geometry that face use of BBR data for EarthCARE's closure assessment.

First, L2-retrieved profiles are ~1 km in diameter, while the BBR was designed to perform best for footprints of ~10 x 10 km. For this configuration, fluxes and radiances computed for sequences of retrieved profiles contribute only ~10% of the signal to each BBR footprint (or pixel). Second, at only ~1 km wide, net horizontal fluxes for each retrieved column, and sequences of them, will rarely be close to zero (e.g., Barker and Li 1997; Marshak et al. 1998). This implies that 3D RT models, as opposed to their ubiquitous 1D counterparts, will be required to make EarthCARE's

radiative closure assessment fruitful. Hence the need for 3D arrays of data that describe the Earth-atmosphere system adjacent to the ~1 km-wide retrieved L2-cross-section.

Fortunately, BBR data are not bound to 10 km resolution, as point-spread function widths of its native radiances are ~0.7 km. This offers much flexibility to the design of the closure assessment (e.g., Tornow et al. 2018). The extreme case is use a single along-track line of BBR radiances that overlap, at best approximately, the ~1 km wide curtain of

L2-retrieved profiles, referred to hereinafter as the L2-plane. This would, however, degrade BBR performance, via reduced signal-to-noise ratio and pointing accuracy, and thus weaken closure assessments. Alternatively, one could attempt an across-track "broadening" of the L2-plane so as to cover as many BBR native radiances as deemed necessary. Regardless of the route taken and the size of domains over which closure assessments are to be performed, there is the ever-present related issue of lateral flow of photons both within assessment domains *and* between assessment

domains and their adjacent areas. Taking these issues together, EarthCARE's science team opted for its closure assessment experiment to use 3D RT models applied to assessment domains centred on the L2-plane with across-track widths appreciably greater than 1 km (Illingworth et al. 2015).



The method for approximating 3D geophysical variables adjacent to the L2-plane, in order to safely use both 3D RT

models and BBR data, is the radiance-matching *Scene Construction Algorithm* (SCA) (Barker et al. 2011), which forms

the basis of the ACM-3D processor. The purpose of the current paper is to recap, in section 2, the SCA, present several

operational details associated with it, and demonstrate, in section 3, its overall performance using simulated observa-

tions for a virtual Earth-observation system (Qu et al. 2023; Donovan et al. 2023). Application of RT models to SCA

products and subsequent radiative closure assessments for the same virtual environments are discussed by Cole et al.

(2023) and Barker et al. (2023). A brief discussion is given in section 4.

## 2. 3D Atmosphere-Surface Scene Construction Algorithm


To begin, EarthCARE's products are partitioned into ~6,565 km long "frames", which makes six frames / orbit. Posi-

tion in a frame is defined by the Joint Standard Grid (JSG). Each frame contains $N_{JSG}$ L2-retreived columns along-

track. Frame widths are 150 km as defined by the MSI's swath; 35 km to the right and 115 km to the left of the L2-

plane (relative to the satellite's direction of motion). Some algorithms require data beyond nadir, and so each frame's

files also contain $n_\varepsilon$ swaths of data from neighbouring frames. JSG coördinates are denoted as $(i, j)$, with $i$ running

along-track from $1 - n_\varepsilon$ to $N_{JSG} + n_\varepsilon$, and $j$ running perpendicular to the L2-plane, which is located at $j = 0$.

As documented by Cole et al. (2023), EarthCARE's radiative products are produced by 1D and 3D broadband RT

models. Some of these products are used to perform radiative closure assessments of L2-retrievals (see Barker et al.

2023). While 1D RT models are applied, in ACM-RT, to all non-corrupt columns in the L2-plane, their results are

averaged, in ACMB-DF, over small domains $D$, referred to hereinafter as *assessment domains*, whose along-track

centres, at $j = 0$, are the L2-plane. Conversely, 3D RT models operate directly on $D$ and domain-average results are

computed in ACM-RT and used in ACMB-DF. The structure of $D$ for $j \neq 0$ is defined by the 3D *Scene Construction*

*Algorithm* (SCA), which is explained in the following subsections. Along-track lengths of $D$, in terms of JSG cells,

are $n_{assess}$ columns(pixels), while their across-track half-widths are $m_{assess}$, making full across-track widths

$2m_{assess} + 1$ columns(pixels). The initial plan is to fix $m_{assess}$ and $n_{assess}$ at 2 and 21, respectively. Thus, $D$ are ~

$5 \times 21$ km, so their areal extents, regardless of location, are ~100 km², which is what the BBR was designed to operate

at (see Illingworth et al. 2015).

The operational SCA is made up of several sub-algorithms. At its core is definition of $D$ at $(i, j \neq 0)$ via MSI radiance-matching (Barker et al. 2011). Other crucial components include: definition of buffer-zones around $D$, as

required by the 3D RT models; screening and ranking of $D$ in an attempt to maximize the usefulness of the radiative closure assessment process; and estimation of errors for TOA fluxes and radiances that stem from the radiance-matching algorithm. To improve readability of the main text, many details of these sub-algorithms are presented in Appendices. General results are shown and discussed in Section 3.

### 2.1. Radiance matching

The core of the SCA, and thus the ACM-3D processor, is passive narrowband radiance-matching of an off-nadir MSI pixel's spectral radiances with their nadir counterparts along the L2-plane (Barker et al. 2011). As this methodology has been described and used elsewhere (Barker et al. 2011; 2012; 2014; Sun et al. 2016), it is recapped briefly here, with details reiterated in Appendix A. Note that all independent variables referred to here are available to the ACM-3D processor from other EarthCARE processors; all of which are reported on in this special issue.

Let $r_k(i, j)$ be MSI radiances, for the $k^{\text{th}}$ channel, where values at position $(i, j = 0)$ align along the L2-plane and have geophysical profiles associated with them. When seeking to populate an off-nadir *recipient* column at $(i, j \neq 0)$ with a suitable *donor* from the L2-plane, the algorithm quantifies how well $r_k(i, j \neq 0)$ match $r_k(m, j = 0)$ for all $m \in [i - M, i + M]$; $M$ being the number of JSG pixels, forward and backward, along the L2-plane one is prepared to allow the algorithm to search. While $M$ could depend on a host of variables, a default value of 200 has been used

thus far. As explained in Appendix A, for a nadir pixel to be a potential donor, the recipient's and donor's surface types (land, sea, and snow/ice) must match, and cosine of solar zenith angles $\mu_0$ and azimuth angles between Sun and satellite tracking direction $\varphi_r$ must differ by less than specified amounts. *Figure 1* shows a schematic of solar-satellite geometry. While the recipient and L2-donor pixels could be required to have the same cloud phase, the intention all along has been for the algorithm to rely just on radiances, not other algorithms.





Of those L2-plane pixels whose MSI radiances best resemble those at $(i, j \neq 0)$, the one lying physically closest to

$(i, j \neq 0)$ becomes the donor, and its profiles of geophysical information get replicated at $(i, j \neq 0)$. This procedure

is performed for all $(i, j \neq 0)$ across the MSI's swath; pixels at $(i, 0)$ donate to themselves. The result is *construction*

of a 3D atmosphere-surface domain made-up of profiles from the L2-plane. Correspondingly, MSI imagery are *recon-*

*structed*, too. *Figure 2* shows a schematic of the radiance-matching procedure. Hereinafter, values at $(i, j)$ that are

based on the SCA are indexed as $(m*(i, j), 0)$, which ties them back to the $m*(i, j)_{\text{th}}$ column/pixel along the L2-

plane.

### 2.2. Assessment domains and their buffer-zones

The 3D RT models used for EarthCARE employ cyclic horizontal boundary conditions. Real cloudy domains and those

produced by the SCA are, however, non-cyclical and so when 3D RT models operate on them, adverse effects near the

perimeter of $D$ are affected by photon paths crossing discontinuous optical properties. One way to deal with this is to

add atmosphere and surface to all edges of $D$ so as to, not eliminate but rather, displace away from $D$ any adverse

effects set-up by the assumed cyclic boundary conditions. Hereinafter, the domain resulting from the combination of

$D$ and its buffer-zones is denoted as $D^+$.

Buffer-zone also accommodate the fact that the BBR's oblique radiances associated with $D$ consist partly of photons

that were scattered by the atmosphere and surface outside of $D$. Barker et al. (2015) described an adjustment to 3D

RT Monte Carlo models that approximates fractional contributions to a BBR radiance from photons whose last scat-

tering event, by any particular scattering species, took place in $D$. Their algorithm, however, is too time-consuming

for EarthCARE's official data processing system. Nevertheless, the buffer-zone should adequately capture contribu-

tions to BBR radiances that come from beyond $D$.

Setting the along- and across-track dimensions of buffer-zones is described in detail in Appendix B. *Figure 3* shows

$D$, its associated $D^+$, and the indices that locate them on the JSG. The number of pixels in the along-track that are

out in front $n_{buffer}^{\searrow}$ and trailing $n_{buffer}^{\swarrow}$ $D$ can vary and depend on the minimum along-track buffer length $d_{\parallel}^{\min}$,





assumed to be ~5 km, BBR viewing zenith angle $\theta_v$, and nearby cloudtop altitudes $h\big(m*(i,j),0\big)$ as defined by

the radiance-matching algorithm. *Figure 4* is a schematic of this process which is detailed in Appendix B.

Note that $n_{buffer}^{\swarrow}$ and $n_{buffer}^{\searrow}$ are independent of $\theta_0$. This is because for EarthCARE's orbit; over 80% of observations

with $\theta_0 < 90°$ have $30° < \varphi_r < 150°$ implying that, for most cases, projection of direct-beam (and hence cloud

shadows) into the along-track walls of $D$ will be small. Exceptions are for very large $\theta_0$ with $\varphi_r < 30°$ or

$\varphi_r > 150°$; but these cases are usually avoided because the RT models rely on the plane-parallel approximation.

Determination of the size of cross-track buffer-zones $m_{buffer}$ depends on $h\big(m*(i,j),0\big)$ adjacent to the sunlit side

of $D$ and minimum size for the across-track buffer-zone $d_{\perp}^{\min}$. If a cloud to the side of $D$ casts a shadow that falls

onto the region formed by $D$ and its along-track buffer-zones, then the cross-track buffer-zone extends to include the

cloud doing the shadowing. Unlike the along-track, $m_{buffer}$ gets applied to both sides of $D$. *Figure 5* summarizes

definition of $m_{buffer}$.

### 2.3. Screening radiative closure assessment domains

Due to computational limitations and time constraints on product production, it is anticipated that only a small number

of $D$ per frame will participate in the 3D RT radiative closure exercise (i.e., processor ACMB-DF). Hence, $D$ that

emerge from the SCA must be "screened and ranked" to ensure that an adequate range of cases, from "simple" to

"complex", be assessed in order to: i) provide well-rounded pictures of algorithmic performances; ii) gauge whether

mission objectives are being met; and iii) provide guidance to data users who wish to focus on select conditions. At

the same time, screening and ranking should be flexible enough to be changed during the mission. For instance, "sim-

ple" scenarios are likely to be of particular interest during the commissioning phase (e.g., mono-phase, single-layer,

overcast clouds).

The screening process eliminates $D$ that are likely to yield uninformative assessments. It has three stages as described

in the following subsections. Each frame has two sets of assessment domains that survive screening. The first set $D_{1D}$





contains domains that 1D RT results are averaged over, while the second set $\mathbf{D}_{3D}$ contains domains that 3D RT models

get applied to. Domains in $\mathbf{D}_{1D}$ and $\mathbf{D}_{3D}$ consist, respectively, of *radiative assessment domains* $D$ and *radiative*

*computation domains* $D^+$ (see *Figure 3*).

### 2.3.1. Screening Stage 1: Failed retrievals and corrupt data

If there are corrupt data or failed retrievals for any column in either $D$ or $D^+$, the domain will not be forwarded to

subsequent processes. This is because they simply cannot be used by 3D RT models. Examples of failed retrievals

include algorithms that did not converge to a solution or converged to values that are out of bounds. Corrupt data, on

the other hand, includes failure at the L1 level and data corrupted during transmission.

### 2.3.2. Screening Stage 2: Geophysical conditions

#### a) Solar zenith angle

Generally speaking, solar RT becomes increasingly complicated as solar zenith angle $\theta_0$ increases. This is because: i)

radiances and fluxes for EarthCARE-sized domains will depend increasingly on conditions outside $D$ and $D^+$ ; and ii)

Earth's sphericity becomes increasingly important; EarthCARE's RT codes are plane-parallel models. Not only do

large $\theta_0$ impact the SCA directly, they stress retrieval algorithms that use MSI data. Hence, $D$ or $D^+$ must have

$$\min\left\{\theta_0\left(i,j\right)\right\} < \theta_0^* \quad \forall\left(i,j\right) \in D \text{ or } D^+, \tag{1}$$

where $\theta_0^*$ is the maximum allowed value of $\theta_0$. Initially, $\theta_0^* = 75°$.

#### b) Single surface type

To focus closure assessments on retrieved cloud and aerosol properties, and reduce potential complications due to

boundary conditions that are outside the purview of EarthCARE's retrievals, $D$ and $D^+$ must have at least 90% of its

area occupied by a single broad class of surface type. These types are: water (oceans and lakes), land, and ice.





*c) Homogeneous land surfaces*

Each JSG pixel has a land-type designation obtained from the International Geosphere–Biosphere Programme (USGS 2018). Let $f_\ell$ be the fraction of JSG pixels in either $D$ or $D^+$ that corresponds to the $\ell^{\text{th}}$ surface-type. For reasons listed above, if $D$ and $D^+$ is a "land" domain, then for them to be included in $\mathsf{D}_{\text{1D}}$ and $\mathsf{D}_{\text{3D}}$, they must have

$$\max\{f_\ell\} > f^*, \tag{2}$$

where $f^*$ is set initially to 0.9.

*d) Surface elevation*

Uncertainties in spectral bidirectional reflectance and emittance functions as well as albedos and emissivities are complicated by variations in surface elevation. In an attempt to limit uncertainties associated with the setting of these lower boundary conditions in the RT models, only very flat assessment domains are allowed. Hence, if $\sigma_{srf}$ is standard

deviation of surface elevation for $D$ or $D^+$, then for them to be included in $\mathsf{D}_{\text{1D}}$ and $\mathsf{D}_{\text{3D}}$, they must have

$$\sigma_{srf} < \sigma_{sfc}^*, \tag{3}$$

where $\sigma_{sfc}^*$ is set initially to 0.1 km.

*2.3.3. Screening Stage 3: Quality of radiance matching*

Where SCA reconstructed MSI radiances are poor, it is reasonable to assume that corresponding constructed 3D do-
mains are, too. Thus, the final screening stage addresses the quality of reconstructed MSI imagery, but it also provides bias-correction estimates for modelled TOA quantities.

For an assessment domain $D$ of $n_{assess}$ JSG pixels along-track and $j \in \left[-m_{assess}, m_{assess}\right]$ across-track, the $n^{\text{th}}$ moment of $r_k$ over $D$, excluding the L2-line along $j = 0$, is



$$\left\langle r_k^n \right\rangle = \frac{1}{2m_{assess}n_{assess}} \sum_{i=i_1}^{i_1+n_{assess}-1} \sum_{j\in[-m_{assess},-1]\cup[1,m_{assess}]} r_k^n(i,j). \tag{4}$$

where $i_1$ is along-track JSG index at the edge of $D$. Corresponding reconstructed values are

$$\left\langle \hat{r}_k^n \right\rangle = \frac{1}{2m_{assess}n_{assess}} \sum_{i=i_1}^{i_1+n_{assess}-1} \sum_{j\in[-m_{assess},-1]\cup[1,m_{assess}]} r_k^n(m*(i,j),0). \tag{5}$$

Therefore, errors stemming from the radiance-matching algorithm, for the $k^{th}$ MSI channel, are

$$\Delta\left\langle \hat{r}_k^n \right\rangle = \left\langle \hat{r}_k^n \right\rangle - \left\langle r_k^n \right\rangle. \tag{6}$$

Let $\left\langle F_{SW} \right\rangle$ and $\left\langle F_{LW} \right\rangle$ be TOA SW and LW fluxes, averaged over $D$, as estimated by angular direction models in

the BMA-FLX processor (Velázquez-Blázquez et al. 2023). Since MSI radiances $r_1$ (0.67 m) and $r_6$ (10.80 m)

often correlate well with $\left\langle F_{SW} \right\rangle$ and $\left\langle F_{LW} \right\rangle$ (Barker et al. 2014), TOA flux bias errors stemming from the radiance-

matching algorithm can be approximated as

$$\Delta\left\langle \hat{F}_{SW} \right\rangle \approx \left\langle F_{SW} \right\rangle \frac{\left\langle r_1 \right\rangle - \left\langle \hat{r}_1 \right\rangle}{\left\langle \hat{r}_1 \right\rangle} \quad \text{and} \quad \Delta\left\langle \hat{F}_{LW} \right\rangle \approx \left\langle F_{LW} \right\rangle \frac{\left\langle r_6 \right\rangle - \left\langle \hat{r}_6 \right\rangle}{\left\langle \hat{r}_6 \right\rangle}. \tag{7}$$

If these values satisfy

$$\left| \Delta\left\langle \hat{F}_{SW} \right\rangle \right| > \Delta F_{SW}^* \mu_0 \quad \text{and} \quad \left| \Delta\left\langle \hat{F}_{LW} \right\rangle \right| > \Delta F_{LW}^*, \tag{8}$$

where $\Delta F_{SW}^*$ and $\Delta F_{LW}^*$ are tolerable broadband TOA flux errors arising from the radiance-matching algorithm, $D$ is

*not* included in neither $D_{1D}$ nor $D_{3D}$. Both $\Delta\left\langle \hat{F}_{SW} \right\rangle$ and $\Delta\left\langle \hat{F}_{LW} \right\rangle$ get passed to ACMB-DF and used to bias-

correct estimated TOA fluxes made by both 1D and 3D RT models. This completes the screening processes leaving

$D_{1D}$ and $D_{3D}$ with $m_{1D}$ and $m_{3D}$ assessment domains, respectively.



**2.4. Ranking radiative assessment domains**

As noted above, the EarthCARE mission will provide near-real time products with limited computational resources. The portion of the processing chain constrained most by this involves 3D SW RT models. For them to achieve adequate signal-to-noise ratios, just a small fraction of the thousands of potential $D^+$ per frame can be assessed. Thus, to enhance efficacy of the radiative assessment process, an algorithm was developed that ranks $D^+$ in $\mathsf{D}_{3D}$. At any time,

ranking can be overruled manually, such as when testing during commissioning phase. The ranking algorithm that was decided upon samples cloud scenarios in proportion to relative frequencies of occurrence.

For the initial version of the algorithm, 1 km resolution MODIS-retrieved values (MYD06_L2) of cloud optical depth $\tau_{cld}$ and cloudtop pressure $p_{cld}$, for 2020, where grouped into 5 x 21 km arrays, to match EarthCARE's planned 5 x 21 km domains $D$, and for each array mean values $\langle \tau_{cld} \rangle$ and $\langle p_{cld} \rangle$ were computed along with total cloud fraction

$A_c$ (Platnick et al. 2015). They were then assigned to bins defined by $10°$ ranges of latitude and longitude, and ranges for $\langle \tau_{cld} \rangle$, $\langle p_{cld} \rangle$, $A_c$, and time of observation of

$$
\begin{array}{lll}
\langle \tau_{cld} \rangle & : & (0,4];(4,23];(23,150] \\
\langle p_{cld} \rangle & : & (0,440];(440,680];(680,\infty) \\
A_c & : & (0,0.25];(0.25,0.75];(0.75,0.99];(0.99,1] \\
time & : & DJF;MAM;JJA;SON
\end{array}
\tag{9}
$$

Since only cloud-bearing domains need be ranked, the number of applicable bins is 23,328, of which the $n^{th}$ has $N_n$ domains. When $\theta_0 \geq 90°$, $\langle \tau_{cld} \rangle$ is not retrieved, so the same ranking procedure uses the remaining three variables

only. Radiative closure assessments are not done when $75° < \theta_0 < 90°$ on account of too many overwhelming uncertainties with retrievals and plane-parallel RT modelling. Thus, domains in this range are not ranked.

For EarthCARE frames with 6,400 1-km L2 columns in the L2-plane there is the potential for 6,400 - 21 + 1 = 6,380 (over-sampled) assessment domains, each of which falls into one of the MODIS bins. Now, using only domains with $A_c > 0.01$, form the frame-specific cumulative frequency such that the $i^{th}$ domain has a value





$$\mathsf{N}\,(i) = \frac{1}{\mathsf{N}\,(6{,}380)} \sum_{j=1}^{i} N_n\,(j) \quad ; \quad i = 1,\ldots,N_{cld} \le 6{,}380.$$ (10)

Then, generate a uniform pseudo-random number $\mathsf{R} \in (0,1]$ and find the closest $\mathsf{N}\,(i)$; this identifies the top-ranked $D^+$. This is repeated, without replacement, until (possibly all) the domains are ranked. The resulting lists are passed on to ACM-RT.

## 3. Results

The SCA's sub-algorithms are discussed in series in this section. The only part not evaluated here is the ranking algorithm (see section 2.4). Its performance is demonstrated in Cole et al. (2023; in this issue) where radiative transfer models act on the highest ranked assessment domains.

### 3.1. Reconstruction of passive radiances

Radiance matching is the essence of the SCA, and a sampling of results is provided here. More details are in Barker et
al. (2011; 2012; 2014) and Sun et al. (2016). The largest impediment facing this algorithm is when conditions off the ground-track (GT) differ much from the relatively few samples along the GT (Barker et al. 2011). *Figure 6* shows a full-width segment of the *Hawaii* frame that contains both catastrophic failure due to inadequate conditions along the GT and excellent performance for the opposite reason. Approximately 50 - 100 km east of the GT, between $9.5°\,\mathrm{N}$ and $13°\,\mathrm{N}$, radiances, especially channel 1's, that are associated with much cloud are severely short-changed on ac-
count of a long stretch of near-cloudless conditions at nadir. To partially remedy this, the algorithm would have to be permitted to search the GT much further than 200 km, as in this example, but in so doing it would run the risk of identifying donor columns associated with increasingly different meteorological conditions. On the other hand, be-tween $5°\,\mathrm{N}$ and $8°\,\mathrm{N}$ performance is extremely good across the entire 150 km wide frame.

*Figure 7* shows mean bias errors (MBE) and root mean square errors (RMSE) for reconstructed MSI channels 1 and 4
as functions of distance, east and west, of GT for the full lengths of the three test frames (see Qu et al. 2023). By definition, MBE and RMSE along GT are zero. For channel 1's reconstructions, MBEs out to $\pm 20$ km are generally smaller than 0.05 W m$^{-2}$ sr$^{-1}$, with smaller errors for cloudless pixels. The negative biases for *Baja* and *Hawaii* frames

out past ~ 5 km from GT stem from clouds along the GT being slightly darker than those elsewhere over the frames

(see *Figure 6*). Likewise, MBEs for channel 4 are generally smaller than 0.5 K. Over the 5 km wide assessment domains

centred on GT, MBEs for both channels are almost negligible relative to the frame-wide average values that are listed

along the base of each plot.

Unlike MBEs, however, RMSE values jump immediately off the GT. In general, they continue to increase with distance

from GT, but it is difficult to say at what distance they become unusable. It is important to note that RMSE values

plotted here get reduced by at least a factor of 10 when for averages over assessment domains.

**3.2. Definition of assessment domain buffer-zones**

The planned initial default values for both $d_{\parallel}^{\min}$ and $d_{\perp}^{\min}$ is 5 km. Thus, for $D$ measuring 5 x 21 km, 3D RT will be

applied to $D^+$ that measure at least ~15 km across-track by ~31 km along-track. *Figure 8* shows sizes of $D^+$ along

the *Hawaii* frame. The most notable feature is that along-track lengths vary much more than across-track lengths. This

is driven by the fixed along-track off-nadir views of the BBR; whenever cloud occurs, especially high cloud, values of

$n_{buffer}^{\searrow}$ and $n_{buffer}^{\swarrow}$ exceed 0. Lengths of $D^+$ maximize between about $3°\,$N and $6°\,$N where mean cloudtop altitudes

$h$ reach ~17 km; this despite cloud fractions for $D^+$ being substantially less than 1.

While across-track buffer sizes $m_{buffer}$ depend on $h$, too, they also depend on $\varphi_r$ and $\theta_0$. Near latitude $10°\,$N, while

$\varphi_r \approx 107°$ (Sun is shining in almost perpendicular to GT), $\theta_0 \approx 30°$ and so lengths of cloud shadows cast perpen-

dicular to GT are small. As such, there is little need to expand the domain across-track, and so $m_{buffer} = 0$ and cross-

track size of $D^+$ equal the default 15 km. Near latitude $21°\,$S, however, where mean $h$ are the same as near $10°\,$N,

clouds are more overcast, but equally important, $\varphi_r \approx 130°$, which is still not far off shining in perpendicular to GT,

and $\theta_0 \approx 50°$. Together these conspire to cast cloud shadows well beyond $D$ resulting in $D^+$ that are slightly larger

than the default.


### 3.3. Estimation of SCA-related bias errors for TOA broadband fluxes

In a manner similar to cloud radiative effects, estimation of errors for TOA broadband radiative fluxes that arise from the SCA process provide a simple, integrated indication of SCA performance. *Figure 9* shows cumulative frequency distributions of $\Delta \left\langle \hat{F}_{\mathrm{SW}} \right\rangle$ and $\Delta \left\langle \hat{F}_{\mathrm{LW}} \right\rangle$ (see (7)), for the *Hawaii* frame. Errors for this frame are the largest of the three. As is often the case, errors tend to be largest for SW fluxes, and increase slightly as assessment domain $A_c$ increases. The latter point basically indicates that the SCA does extremely well in clear-sky conditions; even for the

*Baja* frame that was mostly over variable land.

It is encouraging to see that median errors for both bands are negligible for all conditions. Moreover, as *Figure 9* shows, with ~90% of errors being within $\pm 3$ W m$^{-2}$, and slightly better for the other frames, errors imparted on TOA flux estimates by the SCA will not hinder assessment of EarthCARE's objective of retrieving cloud-aerosol properties well enough as to be able to model, on average, TOA fluxes to within $\pm 10$ W m$^{-2}$.

### 4. Summary and discussion

The EarthCARE satellite mission has set itself a very high bar with its plan to infer cloud and aerosol properties, from its observations, well enough that when used to initialize radiative transfer (RT) models, their estimates of TOA fluxes will differ from observed values by, typically, less than $\pm 10$ W m$^{-2}$. To realize, and gauge the success of, this *radiative closure assessment* it will be necessary to employ, operationally, 3D atmospheric RT models. This by itself will set

EarthCARE apart from its predecessors, which have relied entirely on 1D RT models. The immediate issue facing this plan is that L2-retrievals at nadir are just ~1 km wide, and at this scale net fluxes of lateral flowing radiation, not accounted for by 1D RT models, can be substantial (e.g., Marshak et al. 1998). To justifiably use 3D RT models, however, the atmosphere-surface system has to be defined on both sides of the narrow L2-plane. The point of this paper was to demonstrate EarthCARE's method of expanding its L2-retrievals perpendicular to the L2-plane. At the

core of this *3D scene construction algorithm* (SCA) is a radiance-matching (RM) scheme that uses EarthCARE's MSI passive imagery (cf. Barker et al. 2011; 2012; 2014). There are, however, other aspects of this process that have been documented here for the first time.

Synthetic radiances measured virtually by EarthCARE's MSI for three test frames, which were produced for *end-to-end simulation* of EarthCARE's measurement-processing chain (see Qu et al. 2023), were used here to demonstrate

the performance of the SCA. "Reconstruction" of measured MSI radiances forms the foundation of the SCA. Basically, off-nadir pixels are paired with a nadir pixel whose spectral radiances best match theirs. The 3D domains to be used by the 3D RT models are "constructed" by taking columns of geophysical information associated with matching nadir pixels and inserting them at the off-nadir pixels.

Comparing reconstructed radiances to their observed counterparts provides a straightforward indication of the success

of the SCA. If observed radiances are reconstructed poorly, one has to also assume that correspondingly constructed 3D surface-atmosphere domains are unfit for both 3D RT models and radiative closure assessment. SCA errors flare-up when conditions needed at off-nadir locations are lacking from the nearby the L2-plane. Typical performances of the SCA were shown in Section 3.1, and as expected, generally erode the further recipient pixels are from their donors. This is not much of an issue for radiative assessment domains themselves, for they extend just 2 km from the L2-plane.

Their buffer-zones, however, which were described in section 2.2, can be as much as 10 km from the L2-plane (see *Figure 3*). The reason why assessment domains are so small is because EarthCARE's radiative closure assessment seeks to verify the narrow L2 retrievals, not the SCA; the SCA should help achieve this as invisibly as possible.

Once defined, assessment domains and their buffer-zones get screened (see section 2.3) in an attempt to identify those most likely to realize useful closure assessments. Furthermore, the intersection of EarthCARE's data processing limi-

tations with the computational needs of the 3D solar RT model (Cole et al. 2023), assessment domains must be ranked to ensure that the few that can be assessed per frame amount to a good sampling, over the mission's life, of conditions as they occur over the globe and the year. This process was described in section 2.4.

The simple method of estimating TOA flux bias errors that are likely to arise from the SCA (Barker et al. 2014) was assessed using test frame synthetic observations. Reassuringly, it was shown in Section 3.3 that most TOA flux errors

due to the SCA's imperfect atmospheres, for assessment domains measuring 5 x 21 km, can be expected to be much smaller, and rarely larger, than EarthCARE's stated goal of $\pm 10$ W m$^{-2}$.

To conclude, it is tempting to view the SCA as a very cost-effect (i.e., almost free) *scanning active sensor system* (cf. Illingworth et al. 2018). But one almost always gets what one pays for, and the SCA is, when pushed, no exception. Its performance, especially well-removed from the L2-plane, cannot be expected to rival that of an authentic scanning



active sensor system (Barker et al. 2021); to use it as a full-up replacement would be cavalier. In its current role, however, as purveyor of approximate, ancillary information that both facilitates use of 3D RT models and strengthens verification of L2 retrievals, it appears as though its shortcomings will be tolerable and outweighed by its benefits.

**Acknowledgements**

In the short-term, data used for this study can be obtained from the corresponding author. By the time of publication,

ECCC will have an official data repository from which data used here will be downloadable.

**Data availability**

The EarthCARE Level-2 demonstration products the ACM-3D products discussed in this paper are available from https://doi.org/10.5281/zenodo.7117115 (van Zadelhoff et al., 2022).


**Financial support**

This study is supported by Clouds, Aerosol, Radiation - Development of INtegrated ALgorithms (CARDINAL) for the EarthCARE Mission.






APPENDIX A

**Radiance-matching algorithm**

Let $r_k(i, j)$ be MSI radiance, for the $k^{th}$ MSI channel, at $(i, j)$ on the joint standard grid (JSG) in which $j = 0$ is along

the L2-plane. Following Barker et al. (2011), when seeking to populate a column at $(i, j \neq 0)$ with a suitable proxy

from $(i, j = 0)$, begin by computing, for $K_s$ channels,

$$F(i, j; m) = \sum_{k=1}^{K_s} a_k \Lambda(i, j; m) \left[ \frac{r_k(i, j) - r_k(m, 0)}{\max\left[ r_k(i, j), r_k(m, 0) \right]} \right]^2 \quad ; \quad m \in [i - M, i + M]; j \notin 0, \tag{11}$$

where $(m, 0)$ holds potential *donor* profiles for the *recipient* at $(i, j \neq 0)$, $a_k$ are weights that could depend on

channel but were assumed to equal 1, and $M$ is number of pixels, forward and backward, along the L2-plane to be

searched for a donor. The function $\Lambda(i, j; m)$ is defined as

$$\Lambda(i, j; m) = \begin{cases} 0 & ; \quad \chi = .false. \\ 1 & ; \quad \chi = .true. \end{cases} \tag{12}$$

By requiring

$$\chi = \begin{cases} \text{surface at } (i, j) = \text{surface at } (m, 0) \\ \qquad .and. \\ \left| \mu_0(i, j) - \mu_0(m, 0) \right| < \Delta\mu_0 \\ \qquad .and. \\ \mu_0(i, j) \cdot \mu_0(m, 0) > 0 \\ \qquad .and. \\ \left| \varphi_r(i, j) - \varphi_r(m, 0) \right| < \Delta\varphi_r \\ \qquad .and. \\ m \in [1, N_{JSG}], \end{cases} \tag{13}$$





it controls whether or not a pixel at $(m,0)$ is to be considered as a potential donor (see section 2.3). The upper condition means that surface types (land, sea, and snow/ice) must be the same at $(i,j)$ and $(m,0)$. The next two

conditions mean that at $(i,j)$ and $(m,0)$ cosine of solar zenith angles $\mu_0$ must differ by less than $\Delta\mu_0 = 0.005$,

and the Sun must be up or down at both locations. Next, the difference between solar azimuth angles relative to the satellite's tracking direction $\varphi_r$ must be less than $\Delta\varphi_r = 5°$. The final condition simply means that the search cannot

go past the ends of a frame.

For the $1 \le M' \le 2M+1$ values of $F(i,j;m)$ that get computed, define Euclidean distances between the centres

of $(i,j)$ and $(m,0)$ as

$$L(i,j;m) = \Delta L\left[(i-m)^2 + j^2\right]^{1/2}, \tag{14}$$

where $\Delta L$ is horizontal resolution, which for EarthCARE is ~1 km, and sort $F(i,j;m)$ from smallest to largest (i.e.,

best to worst radiance match), with $L(i,j;m)$ going along passively. Denote the reordered sequences as

$\left\{\hat{F}(i,j;m)\right\}_{m=1}^{M'}$ and $\left\{\hat{L}(i,j;m)\right\}_{m=1}^{M'}$. Finally, the donor for location $(i, j \ne 0)$ is deemed to reside at

$$m^*(i,j) = \underset{m\in[1,M'\cdot f]}{\arg\min}\left\{\hat{L}(i,j;m)\right\} \tag{15}$$

which reads: find $m$ that corresponds to the smallest distance between the recipient at $(i,j)$ and those pixels that

constitute the smallest $100f\%$ of $\hat{F}(i,j;m)$. The tuneable parameter $f$ will be set initially to 0.05, but this could

change (see Barker et al 2011). The procedure outlined here is performed for all $(i, j \ne 0)$ across the MSI's swath,

with pixels at $(i,0)$ donating to themselves.






APPENDIX B

**Calculation of buffer-zone sizes**

*a. Along-track length*

Let $h\left(i*(i,j),0\right)$ be cloudtop altitude at JSG cell $(i,j)$ using the SCA's field. Using these values, define for each

$i$ in a frame the highest cloudtop in across-track swaths of JSG pixels spanning the assessment domains as

$$h_\perp^{\max}(i) = \max\left\{h\left(i*(i,j),0\right)\right\} : j \in \left[-m_{assess}, m_{assess}\right]. \tag{16}$$

The symbols $\perp$ and $\parallel$ indicate across- and along-tracks, respectively. Thus, maximum cloudtop altitude in an assessment domain $D(n)$ is

$$h_D^{\max}(n) = \max\left\{h_\perp^{\max}(i)\right\} : i \in \left[n, n + n_{assess} - 1\right]. \tag{17}$$

Consider first the buffer-zone out in front of $D(n)$; applicable most to the BBR's backward view. The minimum

number of JSG pixels that one need be concerned about searching to see if clouds outside of $D(n)$ obscure part of

$D(n)$ is

$$n_{buffer} = nint\left(\frac{\max\left\{d_\parallel^{\min}, h_D^{\max}(n)\tan\theta_v\right\}}{\langle\Delta x\rangle}\right), \tag{18}$$

where $d_\parallel^{\min}$ is the absolute minimum along-track buffer length, assumed for EarthCARE to be ~5 km, $\theta_v$ is BBR

viewing zenith angle, and $\langle\Delta x\rangle$ is mean length of JSG pixels in the along-track direction. For EarthCARE, $\langle\Delta x\rangle \approx 1$

km. If maximum cloudtop altitude across the entire useable part of the 6,000 km frame is

$$h_F^{\max} = \max\left\{h(i,j)\right\} : i \in \left[m_1 + 1, N_{JSG} - m_2 + 1\right], j \in \left[-m_{assess}, m_{assess}\right], \tag{19}$$





then the absolute maximum number of JSG pixels one need search for outside of $D(n)$ that might obscure part of $D(n)$ is

$$n_{buffer}^{\max} \approx nint\left(\frac{h_F^{\max}\tan\theta_v}{\langle\Delta x\rangle}\right). \tag{20}$$

This can be reduced by knowing the location of the along-track JSG pixel that houses the highest cloud between the last pixel of $D(n)$ and $n_{buffer}^{\max}$ pixels out in front of it. This is expressed as

$$n_{buffer}^{\max}(n) = \underset{i\in\left[n+n_{assess}+n_{buffer}(n),\, n+n_{assess}+n_{buffer}(n)+n_{buffer}^{\max}\right]}{\arg\max} h_{\perp}^{\max}(i). \tag{21}$$

Proceed then to search

$$h_{\perp}^{\max}(i) \geq \frac{\left[i-\left(n+n_{assess}\right)\right]\langle\Delta x\rangle}{\tan\theta_v} \;:\; i = n+n_{assess}+n_{buffer},\ldots,n+n_{assess}+n_{buffer}^{\max}(n), \tag{22}$$

where $n_{buffer}$ comes from (18). Let values of $i$ for which the condition in (22) is met form the set $\{i'\}$. Therefore, the size of the buffer-zone for the back-looking view associated with $D(n)$ is

$$n_{buffer}^{\swarrow}(n) = \begin{cases} n_{buffer} & ; \quad \text{if (22) is never met} \\ \max\{i'\} & ; \quad \text{if (22) is met}. \end{cases} \tag{23}$$

To set the size of the buffer-zone for the fore-looking view $n_{buffer}^{\searrow}(n)$, (21) through (23) are reapplied reversing

indices. *Figure 4* is a schematic of this process.

Note that determination of $n_{buffer}^{\swarrow}(n)$ and $n_{buffer}^{\searrow}(n)$ are independent of $\theta_0$. This is because for EarthCARE's orbit over 80% of observations with $\theta_0 < 90°$ have $30° < \varphi_r < 150°$ implying that, for most cases, projection of direct-





beam (and hence cloud shadows) into the along-track ends of $D(n)$ will be small. The exceptions are for very large

$\theta_0$ with $150° < \varphi_r < 30°$; but most of these cases will be avoided via the screening process as discussed in section

400 2.3.2.

*b. Across-track length*

Next, determine the size of across-track buffer-zones $m_{buffer}$. They are set using cloud information on the sunlit side

of $D(n)$. Here the idea is that if a cloud anywhere to the side of $D$ casts a shadow that falls onto the region formed

by $D$ and its along-track buffer-zones, then the across-track buffer-zone extends to include the cloud doing the shad-

405 owing. Once $m_{buffer}$ is established, it is applied to the opposing side of $D(n)$, too.

Begin by setting a minimum size for the buffer-zone of $d_\perp^{min}$, so that searching starts at cross-track JSG pixel

$\pm m_{assess} \pm m_{buffer}$, where $m_{buffer} = nint\left(d_\perp^{min} / \langle \Delta y \rangle\right)$ is the minimum buffer size in JSG pixels; choice of using $+$

or $-$ depends on which side of $D$ the Sun is on. For EarthCARE, $d_\perp^{min} = 5$ km. One then determines the highest

cloud in the row of JSG pixels at distance *j* pixels from L2-plane as

410 $$h_\parallel(n,j) = \max\left\{h(i,j)\right\} : i \in \left[n - n_{buffer}^{\swarrow}(n), n + n_{assess} + n_{buffer}^{\searrow}(n)\right]. \tag{24}$$

If it happens that

$$h_\parallel(n,j) \geq \frac{abs(j - m_{assess})\langle \Delta y \rangle}{\tan\theta_0 \sin\varphi_r}, \tag{25}$$

is satisfied at *j*, the across-track buffer gets reset to $m_{buffer} \leftarrow abs(j - m_{assess})$. This process is continued out to the

edge of the MSI's swath. *Figure 5* is a schematic of this procedure. The aggregation of an assessment domain *D* and

415 its buffer-zones is denoted as $D^+$.





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

niques*, to be submitted.

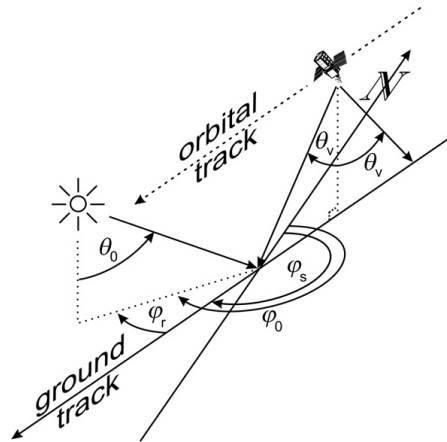

**Figure 1:** Schematic showing solar zenith angle $\theta_0$, solar azimuth angle relative to north $\varphi_0$, satellite-tracking vectors relative to north $\varphi_s$, solar azimuth angle relative to satellite-tracking direction $\varphi_r$, and the BBR's two off-nadir
470                                                     zenith angles $\theta_v$.





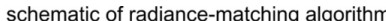

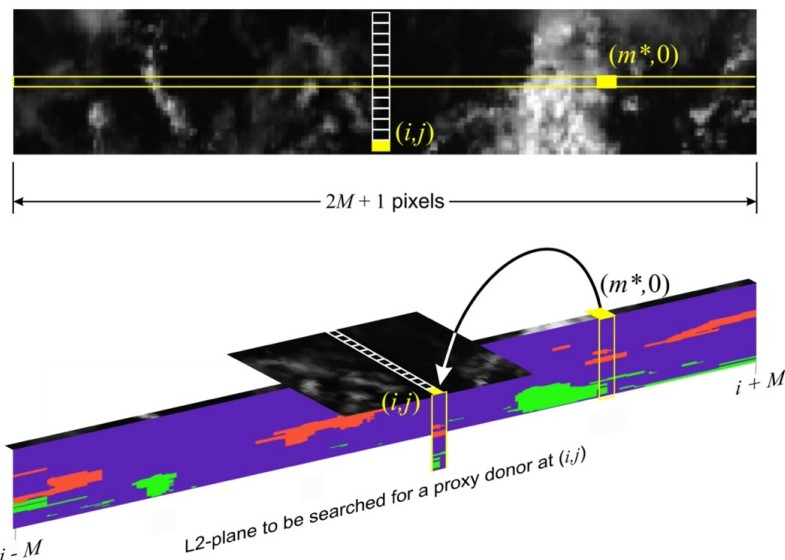

**Figure 2:** Top panel shows a swath of passive imagery and the location of an off-nadir pixel at $(i, j)$ whose radiances match best with those along nadir at $(m*, 0)$. Lower panel is a schematic illustrating attribution of the (recipient) column associated with the pixel at $(i, j)$ to the (donor) column of information inferred from EarthCARE's active-passive measurements at $(m*, 0)$.

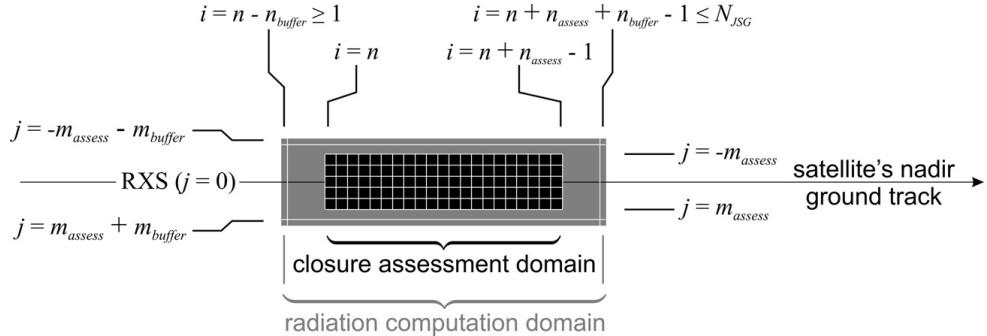

**Figure 3:** Schematic showing the radiative closure assessment domain $D(i)$ (black) and the radiation computation domain $D^+(i)$; the union of $D(i)$ and the buffer-zone (shaded). See the text for definitions of indices.





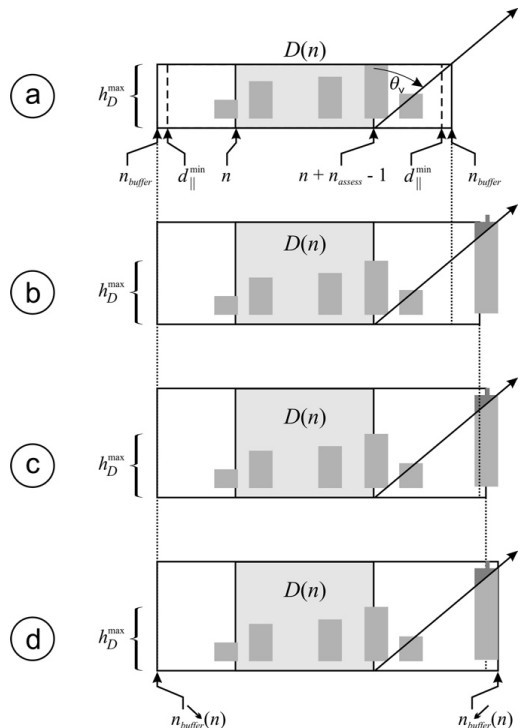

**Figure 4:** Schematic showing the procedure for finding along-track buffer-zone size; in this case $n_{buffer}^{\swarrow}$. (a) $h_D^{max}$ is

maximum cloudtop altitude in assessment domain $D$, and $d_{\parallel}^{min}$ is the smallest size buffer-zone allowed. (b) As the

algorithm searches out in front of $D$, a cloud is encountered; part of which lies between $D$ and the satellite, and so the
buffer-zone increases. (c) Continuing the search, still a higher cloud is encountered between $D$ and the satellite, thus
increasing the buffer-zone further. (d) There is still more cloud between $D$ and the satellite so the buffer-zone in-

creases again, this time to its final value of $n_{buffer}^{\swarrow}$, despite the possibility of cloudtops further out being higher yet

still. See text and Appendix B for details.






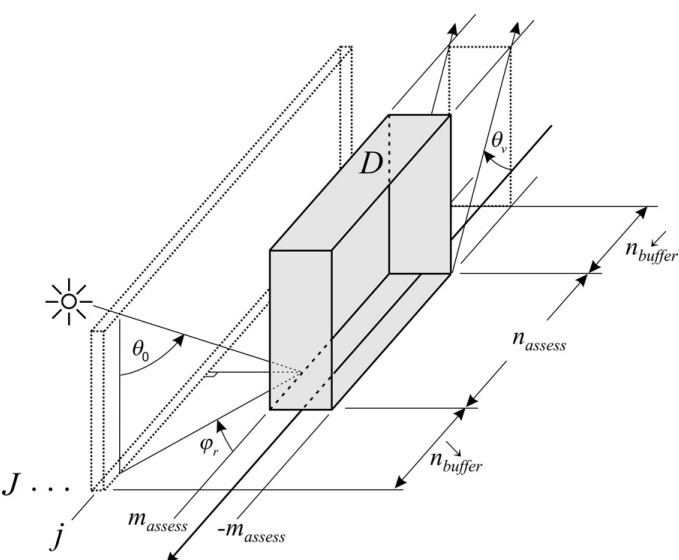

**Figure 5:** Schematic showing the procedure for finding the size of the cross-track buffer-zone. See text and Appendix B for details.

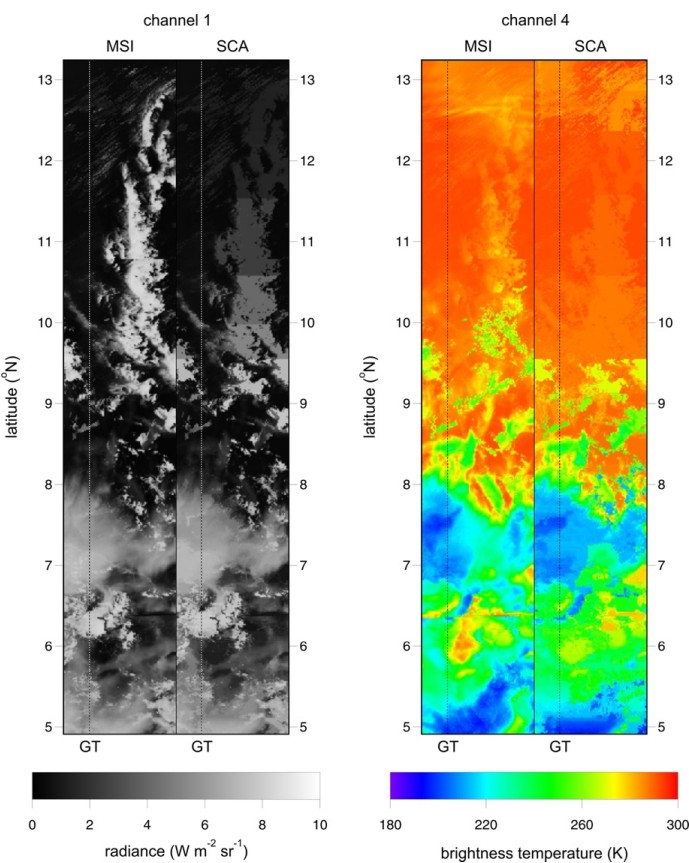


**Figure 6:** Left panel shows MSI and corresponding SCA channel 1 radiances for a segment of the *Hawaii* frame
measuring 150 km wide and 900 km long. Right panel is corresponding brightness temperatures from channel 4 radi-
ances. Dashed lines marked GT indicate the ground-track, which coincides with the L2-plane.

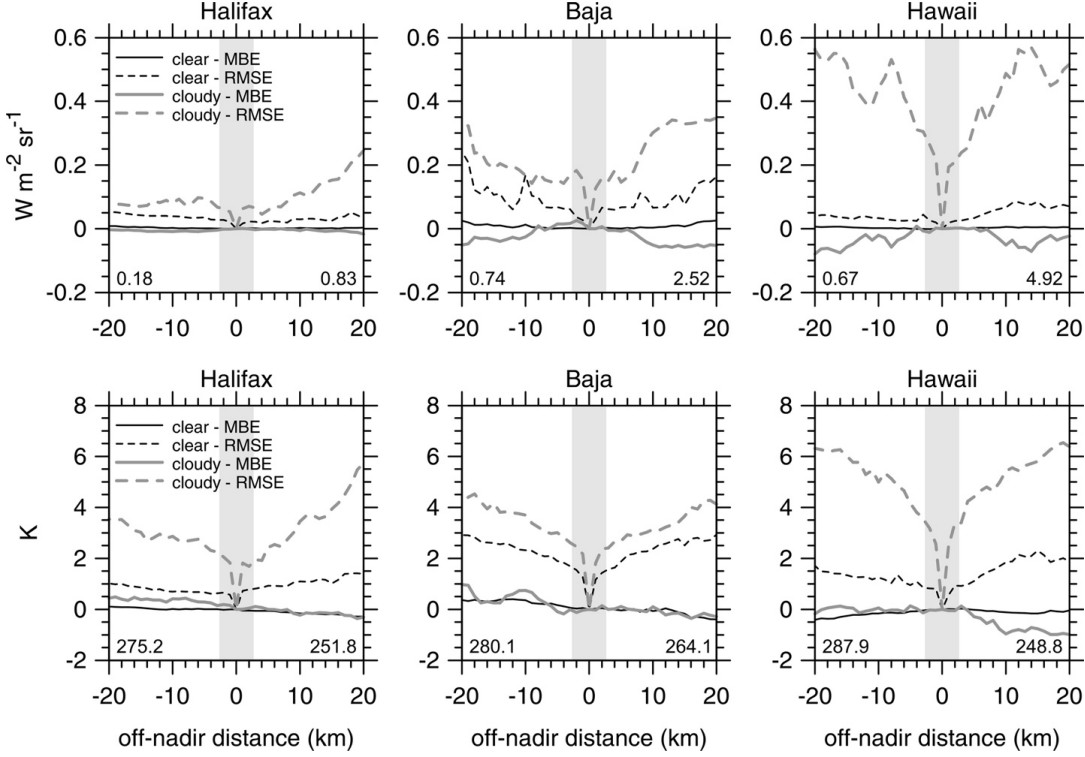


**Figure 7:** Full-frame mean bias errors (MBE) and root mean square errors (RMSE) for SCA reconstructed radiances as functions of distance (-ve values are W) from the ground-track (GT in *Figure 6*). Upper and lower rows are for MSI channel 1 radiances and 4 brightness temperatures. Full-frame mean values are listed on the base of each plot: clear-sky on the left, cloudy-sky on the right. Grey bands indicate EarthCARE's default assessment domains.


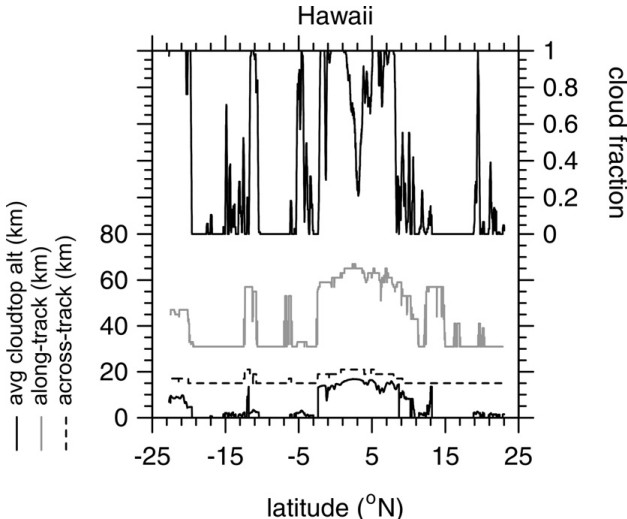

**Figure 8:** Along-track and across-track length of assessment domain plus buffer-zones as functions of latitude for the *Hawaii* frame. All assessment domains are 5 km across-track by 21 km along-track. Minimum size for all buffer-zones is 5 km. Also shown are corresponding values of mean cloudtop altitude and cloud fraction for 5 x 21 km assessment domains.


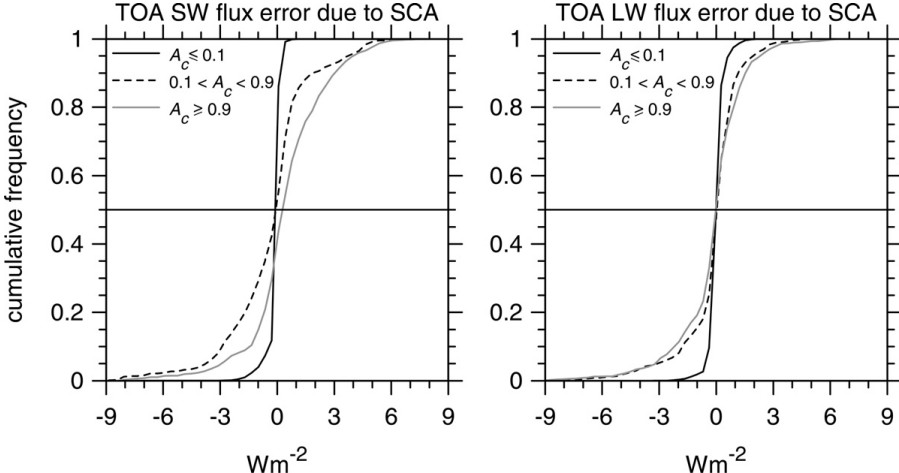

**Figure 9:** Cumulative frequency distributions of estimated errors in broadband TOA upwelling SW and LW fluxes stemming from the SCA for all 5 x 21 km assessment domains in the *Hawaii* frame. Distributions are partitioned for three ranges of assessment domain total cloud fraction $A_c$.
