# Peer review of "Across-track Extension of Retrieved Cloud and Aerosol Properties for the EarthCARE Mission: The ACM-3D Product"

_Atmospheric Measurement Techniques, 2022_

## Author Comment (AC1)

**Answers to Referee #1**

**RC1**: 'Comment on amt-2022-301', Anonymous Referee #1

The authors describe the scene construction algorithm (SCA) for EarthCARE Level-2 data products. The algorithm is based on earlier work by Barker et al. (2011, 2012) that uses spectral radiances to transfer cloud and aerosol vertical profiles derived over ground-track of active sensors to cross-track pixels. The manuscript includes descriptions of three stages of screening process and how to determine buffer zones. The authors define the error due to SCA as the domain averaged radiance difference between observed radiances and transplanted radiances, scaled by the mean flux over the ground-track portion of the domain. The authors show that the error due to SCA is well below 10 Wm-2, which is the error budget of EarthCARE including all algorithms.

The manuscript is well written and easy to follow. I only have minor comments.

**Thank you for the compliment!**

*General comments*

The paper does not discuss what channels/wavelengths are used for the scene construction algorithm and the total number of ks described on line 95. Channels used in the SCA are probably different for day and night. But do they also vary depending on scenes/locations? Some details are discussed in Barker et al. (2011). But it is not obvious from their paper which combination of cannels is used in the actual SCA. Could you include descriptions of channels/wavelengths used in the SCA?

**You are right; we failed to mention what channels are used. They are now mentioned at the beginning of section 3.1. Four channels, i.e. 1 (0.67 μm), 4 (2.21 μm), 5 (8.80 μm), and 7 (12.00 μm), are used by the SCA algorithm during Sun-up periods. At nighttime, however, shortwave channels will not be used. This is stated following (1) (and following (13) in the Appendix). In addition, if a channels' value is zero or missing, it will not be by the SCA either.**

One of conclusions is that the error by the SCA is less than 5 Wm-2 or 3 Wm-2. However, this is based on results using the Hawaii frame (line 276 to 280). Is this also true for other two frames used for testing?

**On line 272 we stated that of the three frames, errors for the Hawaii frame are the largest.**

*Specific comments*

Abstract

It is not described anywhere in the manuscript what four sub-algorithms are.

**We changed the word "sub-algorithm" to "components". Their components are discussed in the 4 subsections in section 2.**

The last sentence of the abstract and Section 3.3.

The direct way to show that the SCA helps more than hinder toward achieving the radiative closure goal of EarthCARE is to show that TOA flux error with and without the SCA. But this study did not address that. The SCA can help reducing the TOA flux error in two says. One way is by identifying clear-sky for the entire BBR footprint. If I look Figure 2 of Ham et al. (2015), nearly 30% of along-track clear-sky scenes contains up to 10% clouds. The SCA should reduce the TOA flux error identifying clouds present off-nadir. Second, the SCA can provide better off-nadir cloud information than no information. Top two plots of Figure 4 of Ham et al. (2015) show improvements of TOA fluxes (smaller differences between CERES-derived and computed fluxes) with the SCA. Also, the left plot of Figure 6 shows the improvement for almost all cloud types. If the authors prefer to estimate in their way, both effects together can be estimated by limiting the area of averaging radiance just over along-track in Eq. (5).

**Your comments are all true, but we're not really concerned with modelling TOA fluxes in this report; that is the focus of a separate paper. For the case at hand, we're never sure how much modelled fluxes, based on SCA clouds, will be in error, but we can estimate bias errors likely to stem from the SCA process by checking known differences between "constructed" and "measured" radiances. Moreover, we tried to emphasize that the SCA is just a tool that helps facilitate radiative closure assessment. Ideally, this "facilitator" does not hinder assessments of the retrievals. When constructed and measured radiances don't (do) agree, we can assume that subsequent modelled broadband fluxes will be a poor (good) estimate of unmeasured real broadband fluxes.**

Section 2.3

The authors describe three stages of screen processes. If a domain contains corrupted data are rejected (line 154), I am wondering what is the fraction of domains that pass this screen process. Do you have an estimate of how often domains are rejected? Could you include the number (yield) based on scenes the authors worked on so far? If active sensor retrievals systematically fail for certain type of clouds, such as deep convective clouds, then these clouds have never been

included in the radiative closure assessment. Could you include author's thoughts/concerns that the closure is performed preferably toward certain cloud types?

**Synthetic data used thus far are basically "perfect". Any corruption would be purposely imposed by us. Moreover, you're correct that if active sensors fail systematically for certain conditions, those conditions will not be assessed. We do not feel comfortable speculating in this paper about active sensor failure rates... those will possibly be addressed in other papers in this special issue.**

One of variables used for screening is the solar zenith angle. Currently, the threshold of the solar zenith angle is 75 degrees. In addition, homogeneous land surface and standard deviation of surface elevations are used to screen scenes to be used for radiative closure. Can the authors estimate the faction of Ds that pass these screening process?

**Screening success rates for the three frames used here are now included in Table 1.**

Line 213. Could you explain what 2020 mean?

**We have now added "inferred from measurements made during 2020".**

Reference

Ham, S.H., S. Kato, H. W. Barker, F. G. Rose, and S. Sun-Mack, Improving the modeling of short-wave radiation through the use of a 3D scene construction algorithm, Q. J. R. Meteorol. Soc., (2015), DOI: 10.1002/qj.2491.

**Citation**: https://doi.org/10.5194/amt-2022-301-RC1

---

## Author Comment (AC2)

**Answers to referee #2**

RC2: 'Comment on amt-2022-301', Anonymous Referee #2

*General comments*

The authors present a simple yet elegant concept for building full three-dimensional atmospheric profiles. The exposition is detailed but very clear (with a few minor comments that I will address shortly). The benefits to the EarthCARE mission are obvious and immediate, yet the authors are not blind to the limitations of the proposed approach (just to give one example, the discussion of Figs. 7 and 9 contains a fair appraisal of the algorithm's estimated error). It is clear that this manuscript does not constitute a full demonstration of the algorithm's performance, but I also appreciate that this was not possible within the scope of this submission, and the authors do include a reference to another contribution (that I have not reviewed). Technically, the paper appears sound, even though I do not feel fully qualified to review all technical details (especially appendix B).

**Thank you for your compliments!**

I would rate the manuscript length as being perhaps on the longish side, but without being excessive.

**Specific comments**

The term "assessment domain" (line 54) and the processor designations ACM-RT and ACMB-DF (lines 74 and 75) have not been formally introduced before first use. However, this may be resolved at the time of publication (I did not check the referenced publications).

**At the beginning of the second paragraph in the Introduction, "assessment domain" is now defined more fully. ACM-RT now has a citation attached to it. Since publication time of the paper that describes and applies EarthCARE's radiative closure assessment is highly uncertain, we just mention that ACMB-DF is, in essence, the radiative closure assessment.**

What is not immediately clear in the main body of the text, is whether the SCA works with one channel at a time, or with a combination of channels. Although this is addressed in the appendix (it is able to use any conceivable, weighted combination), I would suggest explicitly adding this to the main text to improve linear reading, as I believe this influences the interpretation of the results. (Otherwise, I cannot explain the increased error when moving away from the ground track.)

**We mention this now at the beginning of section 3.1.**

Formula (1) uses a minimum, whereas I would have naïvely expected a maximum over the domain, to ensure all values remain bounded by above. Could the authors explain this?

**You're right, it should be max{...}.**

The discussion in 3.1 mention the increasing RMSE with increasing across-track distance from the ground track (visible in Fig. 7). My interpretation is that, with increasing across-track distance, meteorological conditions start to differ more, and it becomes increasingly difficult to match the ensemble of MSI radiances with satisfying accuracy. I think the text would benefit from a brief explanation by the authors.

**What you've described is true, especially out toward the edges of the MSI swath; i.e., up to 130 km off-nadir. But Fig. 7 only goes out +/-20 km so meteorological changes are not too pronounced. Note that $\partial RMSE / \partial x$ , where $x$ is distance from nadir, maximizes for very small $x$. This is "not" due to meteorological changes, but rather the algorithm being supplied with incomplete information from the ranges that are searched along-track. The other side of this issue is searching along-track too far from nadir and running the risk of exactly the issue you raised... hence, the reason why we are likely to limit the search to ~200 km.**

The role of Lambda in Appendix A (line 345) appears to be a selector for suitable donor/acceptor pairs. I take it that any such pairs with F exactly equal to zero disqualify immediately for the selection in formula (15), because they would otherwise falsify the ranking? This is not stated explicitly in the text.

**Good point! We were trying to express the algorithm symbolically in a new way by using lambda (this is not in previous presentations). Lambda is supposed to *exclude* columns from the list of potential donors, not make them the *most appropriate*! Moreover, it can be removed from the summation, too. We think this works... lambda is either 1 (usable) or -1 (unusable). Then, just above (14) it is noted that only $F \geq 0$ are considered.**

Appendix B felt quite technical, and I admit that I could not fully grasp the technical details.

**We agree. It wasn't easy translating what is a fairly simple algorithm/code into text. We felt, however, that it should be presented at least once, and that an Appendix was the appropriate place to do so.**

Figure 5 does not mention or show m_buffer (across-track) or clouds, is this the figure that was intended to be shown?

**You are right, it doesn't. We have redrafted the figure, and now it indicates *m_buffer*.**

In Figure 6, I would suggest adding a marginal (side) plot to the left of the left half of channel 1, plotting channel 1 nadir radiance as a function of latitude (and similarly for channel 4), to assist the reader in assessing the working of the algorithm. For the case shown, I would expect a relatively flat and relatively low nadir radiance profile between ~10 and 13 degrees N, which would explain the gradual failure of the algorithm in matching the high off-nadir radiances, and the visible "banding" in the reconstructed radiances.

**Note that nadir is the line labelled "GT", so there are no errors along it. We have, however, added plots, that we believe the Reviewer is suggesting, that correspond to transects running along the centres of these domains, which are 40 km east of nadir. We consider this to be a distance well beyond where the SCA is expected to be needed most. These plots show closely what the Reviewer described.**

*Suggested technical corrections*

What follows is a list of suggestions that I would humbly propose. Please note that I am not a native speaker, so I respectfully defer to the editor and authors for any final decision.

- 14 (abstract) "out to ~15 km on both sides of nadir": suggest adding "along-track" explicitly for clarity

**Actually, it's "across-track", but you're right in that it did lack clarity.**

- 48-49 possible grammar mistake in "The extreme case is use"

**Right, it should be "... is to use...".**

- 66 typo in "retrieved"

**Got it.**

- 83 possible missing article in "At its core is definition of D"

**We think it is correct, but we modified to "... is the definition...", which we think is still correct.**

- 114-115 possible redundancy in "adverse effects near the perimeter of D are affected by [...]"

**We like what we have.**

- 119 possible grammar mistake in "Buffer-zone also accommodate"

**Should be plural; "Buffer-zones...".**

- 130 semicolon where a comma was expected

- 133 semicolon where a comma was expected

**Right... not sure how semicolons got, and remained, there.**

- 195 symbol (μ) appears to be missing in wavelength units (twice)

**Fixed, but not sure why it didn't come through in the original???**

- 213 possible typo in "where"

**Got it... should have been "were".**

- 214 suggest adding a comma after "and for each array"

**Added.**

- 254 text unclear in "[values] get reduced by at least a factor of 10 when for averages [...]"

**The word "when" has been deleted.**

- 309-310 a verb appears to be missing in the subsentence spanning these 2   lines

   **It was incorrect, and it has been improved.**

- 317 possible typo in "cost-effect"

   **Has been corrected to "... cost-effective...".**